# Deciphering the change in root system architectural traits under limiting and non-limiting phosphorus in Indian bread wheat germplasm

Palaparthi Dharmateja[1], Manjeet Kumar[1], Rakesh Pandey[2], Pranab Kumar Mandal[3], Prashanth Babu[1], Naresh Kumar Bainsla[1], Kiran B. Gaikwad[1], Vipin Tomar[4], Kamre Kranthi kumar[1], Narain Dhar[5], Rihan Ansari[1], Nasreen Saifi[1], Rajbir Yadav[1]*

**1** Division of Genetics, ICAR-Indian Agricultural Research Institute, New Delhi, India, **2** Division of Plant Physiology, ICAR-Indian Agricultural Research Institute, New Delhi, India, **3** ICAR- National Institute for Plant Biotechnology, New Delhi, India, **4** Department of Research and Crop Improvement, Borlaug Institute for South Asia, Ludhiana, Punjab, India, **5** Department of Research and Crop Improvement, Borlaug Institute for South Asia, Jabalpur, Madhya Pradesh, India

* rajbiryadav@yahoo.com

## Abstract

The root system architectures (RSAs) largely decide the phosphorus use efficiency (PUE) of plants by influencing the phosphorus uptake. Very limited information is available on wheat's RSAs and their deciding factors affecting phosphorus uptake efficiency (PupE) due to difficulties in adopting scoring values used for evaluating root traits. Based on our earlier research experience on nitrogen uptake efficiency screening under, hydroponics and soil-filled pot conditions, a comprehensive study on 182 Indian bread wheat genotypes was carried out under hydroponics with limited P (LP) and non-limiting P (NLP) conditions. The findings revealed a significant genetic variation, root traits correlation, and moderate to high heritability for RSAs traits namely primary root length (PRL), total root length (TRL), total root surface area (TSA), root average diameter (RAD), total root volume (TRV), total root tips (TRT) and total root forks (TRF). In LP, the expressions of TRL, TRV, TSA, TRT and TRF were enhanced while PRL and RAD were diminished. An almost similar pattern of correlations among the RSAs was also observed in both conditions except for RAD. RAD exhibited significant negative correlations with PRL, TRL, TSA, TRT and TRF under LP (r = -0.45, r = -0.35, r = -0.16, r = -0.30, and r = -0.28 respectively). The subclass of TRL, TSA, TRV and TRT representing the 0–0.5 mm diameter had a higher root distribution percentage in LP than NLP. Comparatively wide range of H' value i.e. 0.43 to 0.97 in LP than NLP indicates that expression pattern of these traits are highly influenced by the level of P. In which, RAD (0.43) expression was reduced in LP, and expressions of TRF (0.91) and TSA (0.97) were significantly enhanced. The principal component analysis for grouping of traits and genotypes over LP and NLP revealed a high PC1 score indicating the presence of non-crossover interactions. Based on the comprehensive P response index value (CPRI value), the top five highly P efficient wheat genotypes namely BW 181, BW 103, BW 104, BW 143 and BW 66, were identified. Considering the future need for developing resource-efficient

**Data Availability Statement:** All relevant data are within the manuscript and its Supporting Information files.

**Funding:** The financial or material support was provided by ICAR-Indian Agricultural Research Institute, New Delhi.

**Competing interests:** No conflict of interest.

wheat varieties, these genotypes would serve as valuable genetic sources for improving P efficiency in wheat cultivars. This set of genotypes would also help in understanding the genetic architecture of a complex trait like P use efficiency.

## Introduction

Wheat (*Triticum aestivum* L.) is a major staple food crop across the globe that contributes to one-fourth of total dietary protein and one-fifth of calorie intake. In India, recently (2019–20), wheat has recorded the all-time highest production, estimated at 107.59 million tonnes [1]. Since the inception of the green revolution era (the 1960s), productivity has continuously increased with the combined effects of responsive genotypes and agronomic interventions including fertilizer management. However, currently, considering environmental, health concerns and factor productivity (production per unit input), the focus is shifted to resource/fertilizer use efficient genotypes in accordance to the policy and market demand.

Among the three major organic minerals crucial for plant's survival and growth, Phosphorus (P) is a crucial element, which plays a vital role in energy, sugar metabolism, enzymatic reaction, and photosynthesis. P is also a component of nucleic acid, plant hormones, and largely defines crop quality and yield [2, 3]. P is mainly absorbed by the plants in the form of phosphate; however, because of its high reactivity with some metal ions in the soil like $Ca^{2+}$ in alkaline soil; and $Fe^{3+}$, $Mn^{2+}$ and $Al^{2+}$ in acidic soil, it naturally gets precipitated, resulting in low availability of P to the plants [4]. Besides, organic material present in the soil can also bind to P usually in phytate form (inositol compounds) [5], which again hampers its uptake through plant roots. Although excess P application generally results in P toxicity in plants leading to the delayed formation of reproductive organs [6], it also leads to micronutrient deficiencies resulting in reduced yield.

The amount of unabsorbed and unutilized P would flow with the rain or irrigation water into the river resulting in eutrophication [7], a process of making water bodies rich in nutrients and hence leading to the profuse growth of plants. Globally P reserves are declining and will be depleted by the end of this century [8]. In contrast to nitrogen which can be fixed from the atmosphere, weathering minerals are the lone non-renewable natural source of P [9]. P fertiliser's primary source is rock phosphate, which is minable in only a few places in the world [5]. Therefore, concerns have been expressed regarding the potential global scarcity of P in the coming years. With this absolute shortage of P fertilisers, agricultural soils will be depleted by between 4–19 $kg\,ha^{-1}\,yr^{-1}$ worldwide, with an average loss to the tune of 50% of total P losses due to soil erosion [10]. Morocco, a single country, possesses 85% of known remaining deposits. Expected future scarcity is already reflected in the move that the US and China have stopped export for strategic reasons. India depends entirely on imported P fertilizer [5]. With around 24–25 million tonnes of annual $P_2O_5$ requirement for agriculture crop production in India, depleting reserve and high mining cost and analysis of 10–20% use efficiency of applied P is likely to cause massive burden to the Indian economy. The monetary loss of $P_2O_5$ was calculated to the tune of INR 7.81 billion [11].

P is considered one of the important yield-limiting factors in the context of sustainable agricultural production systems in subtropical and tropical conditions [12]. P utilization rate is low (10.7%) in wheat compared to rice (13.1%) and maize (11%) [13], which has demanded the immediate attention of wheat researchers [14] across the globe. In India, 49.3% of soil has a low P, 48.8% is of average P and only about 1.9% are in the high P category [15]. In this

context, genetic interventions are compulsory and need of the hour for breeding P use efficient crops [16]. This is very much essential to make sure of increased productivity and to meet the current food demand, but in ways that are environmentally safe, economically viable, and socially sustainable.

The root system is a vital organ for water and nutrients uptake and it helps in the anchoring of plants to the ground. The physiological characteristics of the root system largely determine the rates of absorption of nutrients and water [17] in crop plants. Plants may exhibit a wide range of adaptations in response to P deficiency, and one of those is root system architectural traits modification [18, 19]. In wheat P requirement during early stages of growth is vital for determining the final yield. Even though sufficient P supplied thereafter, P deprivation during the early developing stages produces significant reductions in tiller production and head development [20, 21]. Plants can modify their roots in response to P stress by stimulating the root hairs [22] and enhancement of lateral root growth [23]. In arabidopsis, P deprivation stimulates the production of a highly branched root system at the expense of the primary root, as evidenced by the increased creation and appearance of lateral roots and root hairs [24, 25]. In other crops like maize and rice, the low P availability affects growth of primary root length, root angle, branching of roots, quantity and length of lateral roots, augmentation of root hair and cluster root formation [26, 27]. In rice PSTOL gene (OsPSTOL) identified their role in phosphorus stress tolerance, change in root system architecture along with improvement of agronomically important traits [28]. Overexpression of wheat homologue of the rice PSTOL gene (OsPSTOL) revealed a substantial influence on root biomass, effective tiller number, flowering time and seed yield, all of which were correlated with TaPSTOL expression [29]. Under low P condition, RSA traits such as root length, root surface area and root volume were moderately heritable in maize [30]. More precisely in wheat developing varieties with the improved inherent ability to absorb more P and understanding the key traits deciding its efficient utilization can be one way to address the problem sustainably. However, before harping on such an adventure, it is essential to have information about the plant traits involved in higher uptake and utilization, the level of variability in the available germplasm, their inheritance, and the way to select for and against these traits to achieve a better genetic gain with low P application. Unfortunately, most of the root system traits influencing P uptake are very difficult to score with different values and hence, tough to characterise different genotypes contrasting for P use response in crop breeding programs. Therefore, not much work has been done globally.

In India except for some isolated effort, none of the Indian wheat breeding programs has any generated information on RSAs which can be integrated into developing P use efficient wheat genotypes. More P efficient cultivars can either reduce P applications or reduce the environmental risk associated with the high P use in agriculture. Hence, the current study was designed to characterize 182 diverse wheat genotypes from the Indian wheat breeding programme for P uptake and understand prominent root system traits largely influencing P use efficiency. The core objective of this work was to identify the P- responsive and P- efficient genotypes and to upgrade the knowledge of the key traits involved in higher uptake and utilization of P.

## Material and methods

### Plant material

Root system architectures (RSAs) study was conducted using 182 improved breeding lines (S1 Table) representing different pedigree groups with diverse parents in their lineage under limited and non-limited phosphorus conditions. Based on our earlier field evaluations, the tested

genotypes are high-yielding and possess resistance to major wheat diseases. The selected genotypes are also in the final stage of evaluation under our Institute trials and few of them better performing may be pipelined to National Initial Varietal Trials (NIVT) in the coming years. In addition, we have also used three popular, high yielding commercial varieties viz., HD2967, HD3226 and HDCSW18 for the estimation of comprehensive phosphorous response.

## Growth conditions

This experiment was conducted in controlled hydroponics conditions of the National Phytotron Facility situated at the Indian Agricultural Research Institute, New Delhi, India during 2019–20. The standard growing conditions with 12˚C—22˚C temperature, 10 h of photoperiod and 70% relative humidity were maintained throughout the growing period. Initially, uniform plump seeds of each genotype were sterilized with 1% sodium hypochlorite for 2 minutes and thoroughly washed with distilled water and wrapped in seed germination paper. Then 5 days old seedlings of uniform size were transferred to the modified Hoagland solution [31]. A hydroponic system was developed from plastic boxes of 18-litre capacity covering with a ceramic lid. Holes of around 8 mm diameter were drilled on the lids. The containers and tops for hydroponic culture were opaque to produce healthy roots and discourage algal growth. The seedlings wrapped in cotton plug were placed in each hole of the lid so that their roots remained immersed in the hydroponic solutions tank.

The basal nutrient solution for hydroponics experiment was comprised of $(NH_4)_2SO_4 \cdot H_2O$ (1 mmol/L), $Ca(NO_3)_2 \cdot 4H_2O$ (1 mmol/L), $KCl$ (1.8 mmol/L), $MgSO_4 \cdot 7H_2O$ (0.5 mmol/L), $CaCl_2$ (1.5 mmol/L), $H_3BO_3$ (1 μmol/L), $CuSO_4 \cdot 5H_2O$ (0.5 μmol/L), $ZnSO_4 \cdot 7H_2O$ (1 μmol/L), $MnSO_4 \cdot H_2O$ (1 μmol/L), FeEDTA (100 μmol/L), $(NH_4)_6Mo_7O_{24} \cdot 4H_2O$ (0.1 μmol/L), and two levels of P were maintained using $KH_2PO_4$ as Non limiting P (0.2 mmol/L) and limiting P (0.02 mmol/L) [31, 32]. The pH of the solution was maintained to 6–6.5 by using 1M HCl and1M KOH and was continuously aerated with the aquarium air pump. Each genotype was replicated three times, and the nutrient solution was replaced with the fresh solution every four days [33] to maintain the continuous supply of nutrients.

## Root analysis

The complete root system of thirty days old seedlings of each genotype across the three replications raised in NLP and LP, were separated from shoot and individual plants were spread out in a tray for scanning. The root system was scanned with Epson professional scanner, and obtained images were analysed with WinRhizo [34] (Pro version 2016a; Regent Instrument Inc., Quebec, Canada). Roots were placed in an acrylic tray with double distilled water root parameters namely TRL, TSA, TRV, TRT, and TRF were recorded by scanning the complete root system. The metric scale was used to measure primary root length manually. The additional pieces of information were generated with the WinRhizo software in which each root parameter namely TRL, TSA, TRV and TRT was divided into five subclasses based on root diameter intervals of 0–0.5 mm, 0.5–1.0 mm, 1.0–1.5 mm, 1.5–2.0 mm and >2.0 mm [35]. The notations of each subclass were given like RL[1-5] for TRL; SA[1-5] for TSA; RV[1-5] for TRV, and RT[1-5] for TRT.

## Statistical analysis

The information on descriptive statistics viz., mean, standard deviation, coefficient of variation, heritability and analysis of variance were generated for traits under study using R-software based STAR (Statistical Tool for Agricultural Research) 2.1.0 software [36]. The estimated values of each genotype's phenotype for further analysis were calculated based on

the best linear unbiased prediction (BLUP) [37]. Based on their performance genotypes are classified into three different groups namely: (i) low performing genotypes ($\leq \bar{x}$ - SD), (ii) medium performing genotypes ($\geq \bar{x}$ - SD) to ($\leq \bar{x}$ + SD), and (iii) high performing genotypes ($\geq \bar{x}$ + SD), where $\bar{x}$ and SD represents mean and standard deviation of particular root traits [38]. The Shannon-Weaver diversity index (H') was calculated for all RSA traits [39, 40].

**Root trait associations and genotypes differential response.** Pearson's correlation coefficient and principal component analysis were performed to understand the associations among the root traits and responsive behaviour of genotypes under contrasting P regimes by using R package version 4.0.1. GGE biplot analysis was also carried out to select the best genotypes for RSAs under both NLP and LP regimes. The P response coefficient (PRC) was calculated as the ratio of the data derived from the LP and NLP treatment of the same genotype for each trait using the following equation, $PRC_{ij} = X_{ij}LP/X_{ij}NLP$. Where, $PRC_{ij}$ is the P response coefficient of the trait ($j$) for the cultivar ($i$); $X_{ij}LP$ and $X_{ij}NLP$ are the value of the root trait ($j$) for the cultivar ($i$) evaluated under LP and NLP, respectively.

**Comprehensive P response.** A comprehensive P response index value (CPRI value) was used to estimate the P stress response capability of all tested wheat genotypes. Fuzzy subordination method could be used to analyse the response of genotypes entirely and avoid the shortage of the single index. The membership function of a fuzzy set is a generalization of the indicator function in classical sets; it represents the degree of truth as an extension of valuation. $U_{ij}$ stands for the membership function value of P efficiency (MFVP) that indicates a positive correlation between the trait and P response. The CPRI value was calculated across traits to evaluate wheat P stress response by using the formula [41, 42] described below.

$$Uij = \frac{PRCij - PRCjmin}{PRCjmax - PRCjmin} \quad (J = 1, 2, 3 \ldots n)$$

Where $U_{ij}$ is the membership function value of the trait ($j$) for the cultivar ($i$) for P response; $PRCjmax$ is the maximum value of the P response coefficient for the trait ($j$); $PRCjmin$ is the minimum value of $PRCj$. Comprehensive P response index value was calculated using the formula:

$$CPRI = \sum_{j=1}^{n} \left[ Uij \times |PRCij| / \sum_{j=1}^{n} |PRCij| \right] \quad (j = 1, 2, 3 \ldots .n)$$

Where CPRI is the comprehensive P response index of each wheat genotype under LP.

## Results

### Measure of variability and heritability of root traits

For seven traits (PRL, TRL, TSA, RAD, TRV, TRT, and TRF) tested under NLP and LP conditions; analysis of variance indicated highly significant variation among the genotypes, phosphorus regimes and Genotype (G) vs Phosphorous (P) interactions for RSAs (Table 1). The highly significant interaction between G × P showed that the genotypes responded differently under contrasting P regimes for the RSAs indicating most of these traits were influenced mainly by genotypic effect of wheat lines followed by NLP or LP phosphorous and their interaction. The coefficient of variation (CV) for the RSAs varied from RAD (5.93%) to TRT (23.68%). As a strong determinant to response to selection, a relatively high broad-sense heritability was observed in TSA (0.83%) followed by TRV (0.82%) and the lowest was obtained in TRT (0.47%). The subclasses of TRL, TSA, TRV and TRF were also having significant G, P and G × P interactions.

**Table 1. Variance components, expected mean squares and heritability for root traits among 182 genotypes evaluated under non-limiting and limiting phosphorus conditions.**

| Variables | | Mean squares | | | CV (%) | Heritability |
|---|---|---|---|---|---|---|
| | | Genotypes(G) | Phosphorus(P) | G x P | | |
| Df | | 181 | 1 | 181 | | |
| PRL | | 205.79(<0.001) | 3450.85(<0.001) | 72.44(<0.001) | 10.12 | 0.64 |
| TRL | | 624689.91(<0.001) | 32037168.18(<0.001) | 112217.76(<0.001) | 16.33 | 0.80 |
| TSA | | 5510.73(<0.001) | 166542.98(<0.001) | 903.00(<0.001) | 15.42 | 0.83 |
| RAD | | 0.0064(<0.001) | 0.1439(<0.001) | 0.0015(<0.001) | 5.93 | 0.76 |
| TRV | | 0.3383(<0.001) | 4.54(<0.001) | 0.058 (<0.001) | 16.34 | 0.82 |
| TRT | | 8912636.24(<0.001) | 43854974.88(<0.001) | 4716964.59(<0.001) | 23.68 | 0.47 |
| TRF | | 12642180.35(<0.001) | 731563841.05(<0.001) | 2471231.18(<0.001) | 21.20 | 0.81 |
| TRL | $RL^1$ | 451361.79(<0.001) | 31206255.42(<0.001) | 93568.30(<0.001) | 19.67 | 0.79 |
| | $RL^2$ | 13762.44(<0.001) | 1601.04(0.124) | 3501.27(<0.001) | 22.46 | 0.74 |
| | $RL^3$ | 143.03 (<0.001) | 580.082(<0.001) | 52.04(<0.001) | 38.20 | 0.63 |
| | $RL^4$ | 20.19(<0.001) | 84.36(<0.001) | 10.65(<0.001) | 58.04 | 0.47 |
| | $RL^5$ | 1.36(<0.001) | 2.85(0.0017) | 0.83(<0.001) | 71.44 | 0.37 |
| TSA | $SA^1$ | 1315.46(<0.001) | 101181.97(<0.001) | 273.49(<0.001) | 18.60 | 0.78 |
| | $SA^2$ | 576.10(<0.001) | 80.18(0.0936) | 153.91(<0.001) | 22.78 | 0.73 |
| | $SA^3$ | 20.35(<0.001) | 79.10(<0.001) | 7.35(<0.001) | 38.57 | 0.63 |
| | $SA^4$ | 5.68(<0.001) | 25.71(<0.001) | 2.90(<0.001) | 58.23 | 0.48 |
| | $SA^5$ | 0.65(<0.001) | 1.43(0.0015) | 0.397(<0.001) | 71.86 | 0.37 |
| TRV | $RV^1$ | 0.04(<0.001) | 2.99(<0.001) | 0.009(<0.001) | 18.13 | 0.77 |
| | $RV^2$ | 0.16(<0.001) | 0.032(0.0484) | 0.046(<0.001) | 23.30 | 0.71 |
| | $RV^3$ | 0.0189(<0.001) | 0.0697(<0.001) | 0.0068(<0.001) | 39.08 | 0.63 |
| | $RV^4$ | 0.0103(<0.001) | 0.0502(<0.001) | 0.0051(<0.001) | 58.48 | 0.49 |
| | $RV^5$ | 0.002(<0.001) | 0.0046(0.0013) | 0.0012(<0.001) | 72.45 | 0.38 |
| TRT | $RT^1$ | 8626618.07(<0.001) | 46922438.02(<0.001) | 4738661.23(<0.001) | 25.17 | 0.45 |
| | $RT^2$ | 154.89(<0.001) | 20.3306(0.2932) | 85.74(<0.001) | 39.69 | 0.44 |
| | $RT^3$ | 6.0091(<0.001) | 15.96(0.0008) | 5.33(<0.001) | 74.45 | 0.11 |
| | $RT^4$ | 1.9639(<0.001) | 3.187(0.012) | 1.6389(<0.001) | 81.34 | 0.16 |
| | $RT^5$ | 0.4998(<0.001) | 1.9377(0.0005) | 0.3613(<0.001) | 88.89 | 0.26 |

## Genotypes response with phosphorus availability (phosphorus response coefficient)

The study of 182 diverse wheat genotypes showed a wide variation in the mean values under NLP and LP for traits under study (Table 2). The independent t-test showed that the mean values of TRL, TSA, TRV, TRT and TRF in LP were significantly high relative to the NLP. Whereas PRL and RAD are substantially higher in the case of NLP compare to LP. The boxplots of these traits over the P regimes have been demonstrated in (Fig 1) with mean values as black "∗". The spread of variations between the $I^{st}$ and $III^{rd}$ quartile were recorded higher for TRL and TRF in LP, which means the level of P influence the expression of these traits in a higher magnitude. Almost all subclasses of TRL, TSA, TRV, and TRT representing various diameters had higher mean values in LP except $RT^3$ was having a high mean value in NLP. TRL and TRF showed high mean values for the phosphorus response coefficient (PRC), followed by TSA, TRV and TRT. PRL and RAD showed a value of PRC < 1.0, suggesting that it was reduced under limited P. The subclasses of TRL, TSA, TRV, TRT also had the >1 PRC in LP which resembles their respective major class except for the $RT^3$.

**Table 2. Measures of variability and Phosphorus Response-Coefficient (PRC) for roots system architectural and their subclass traits under non-limiting and limiting phosphorus conditions.**

| Variable | | Min | | Max | | Mean | | PRC(MEAN) |
|---|---|---|---|---|---|---|---|---|
| | | NLP | LP | NLP | LP | NLP | LP | |
| PRL | | 23 | 14.5 | 63 | 67.3 | 42.46 | 38.9 | 0.92 |
| TRL | | 276.16 | 238.75 | 1623.03 | 2195.67 | 722.78 | 1065.35 | 1.47 |
| TSA | | 25.95 | 26.01 | 159.35 | 226.97 | 72.93 | 97.63 | 1.34 |
| RAD | | 0.24 | 0.22 | 0.45 | 0.52 | 0.32 | 0.30 | 0.94 |
| TRV | | 0.17 | 0.17 | 1.30 | 1.87 | 0.59 | 0.72 | 1.22 |
| TRT | | 587 | 498 | 10509 | 10942 | 2739.39 | 3140.19 | 1.15 |
| TRF | | 754 | 631 | 8837 | 10223 | 2634.84 | 4271.83 | 1.62 |
| TRL | RL[1] | 199.84 | 132.81 | 1602.09 | 1883.86 | 586.41 | 924.50 | 1.58 |
| | RL[2] | 12.91 | 13.50 | 309.26 | 267.64 | 114.62 | 117.05 | 1.02 |
| | RL[3] | 0.24 | 0.10 | 35.41 | 31.50 | 7.96 | 9.41 | 1.18 |
| | RL[4] | 0 | 0 | 14.77 | 23.54 | 2.29 | 2.85 | 1.24 |
| | RL[5] | 0 | 0 | 4.79 | 6.27 | 0.70 | 0.80 | 1.14 |
| TSA | SA[1] | 12.13 | 10.50 | 88.73 | 110.59 | 34.38 | 53.63 | 1.56 |
| | SA[2] | 2.46 | 2.64 | 63.49 | 55.6 | 23.15 | 23.69 | 1.02 |
| | SA[3] | 0.08 | 0.04 | 13.11 | 11.8 | 2.97 | 3.51 | 1.18 |
| | SA[4] | 0 | 0 | 8.11 | 12.41 | 1.22 | 1.52 | 1.25 |
| | SA[5] | 0 | 0 | 3.38 | 4.39 | 0.49 | 0.56 | 1.14 |
| TRV | RV[1] | 0.08 | 0.08 | 0.52 | 0.66 | 0.22 | 0.32 | 1.45 |
| | RV[2] | 0.04 | 0.04 | 1.06 | 0.96 | 0.38 | 0.39 | 1.03 |
| | RV[3] | 0 | 0 | 0.39 | 0.36 | 0.09 | 0.11 | 1.22 |
| | RV[4] | 0 | 0 | 0.36 | 0.52 | 0.05 | 0.07 | 1.40 |
| | RV[5] | 0 | 0 | 0.19 | 0.25 | 0.03 | 0.03 | 1.00 |
| TRT | RT[1] | 584 | 304 | 10477 | 10863 | 2694.31 | 3108.89 | 1.15 |
| | RT[2] | 0 | 0 | 48 | 67 | 10.66 | 10.94 | 1.03 |
| | RT[3] | 0 | 0 | 10 | 9 | 1.52 | 1.28 | 0.84 |
| | RT[4] | 0 | 0 | 7 | 10 | 0.57 | 0.68 | 1.19 |
| | RT[5] | 0 | 0 | 3 | 3 | 0.2 | 0.28 | 1.40 |

## Association studies for RSA traits

Pearson's correlation coefficients revealed a substantial and positive correlations ($p<0.01$ and $p<0.001$) among the measures of root traits (Fig 2). In NLP, a very high positive correlations ($r = >0.80$) were observed for total root length with TSA, TRV and TRF ($r = 0.98$, $r = 0.91$and $r = 0.90$ respectively); and for total root surface area with TRV and TRF ($r = 0.98$ and $r = 0.86$). Similarly, moderate correlations ($r = 0.61$–$0.80$) were observed for primary root length with TRL, TSA and TRV ($r = 0.63$, $r = 0.64$ and $r = 0.62$); for total root tips with TSA and TRL ($r = 0.65$ and $r = 0.71$); and for total root fork with TRT and TRV ($r = 0.64$ and $r = 0.78$). RAD had only positive significant correlation with TRV ($r = 0.43$) and TSA ($r = 0.24$), and non-significant correlation with all the remaining root traits. The other correlations among the RSAs were found to be weak or non-significant. An almost similar pattern of correlations among the RSAs was also observed under LP conditions (Fig 2). However, it is interesting to see a significant negative correlations of root average diameter with PRL ($r = -0.45$), TRL ($r = -0.35$), TSA ($r = -0.16$), TRT ($r = -0.30$), TRF ($r = -0.28$) and non-significant positive association with TRV ($r = 0.09$).

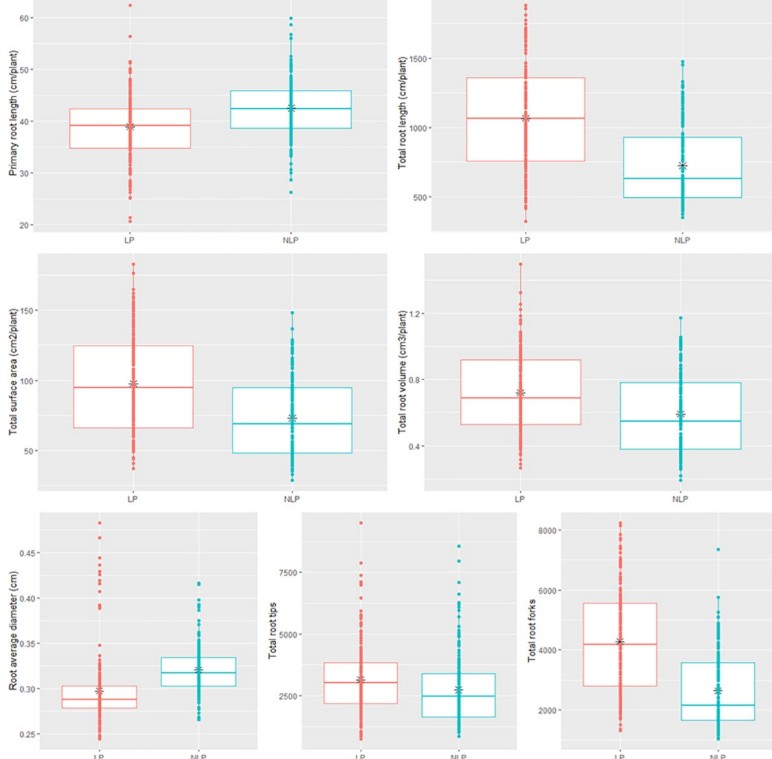

**Fig 1. Box plot showing the median and range of phenotypic variation in wheat genotypes in non-limiting and limiting phosphorus conditions.**

### Fine root distribution patterns under differential P regimes

The distribution of fine roots over various root diameter intervals in both P was analysed to understand the differential relationship that exists between root diameter and other root characteristics (Table 3). The subclasses of TRL, TSA, and TRT representing the diameter of 0–0.5 mm viz., RL$^1$, SA$^1$, and RT$^1$ had higher root distribution percentages than other diameter classes across the genotypes under both P regimes. However, RV$^2$ (a subclass of TRV representing 0.5–1.0 mm diameter) had a higher root percentage than other subclasses of TRV. The subclass of TRL, TSA, TRV and TRT representing the 0–0.5 mm diameter had a higher root distribution percentage in LP than NLP. While the subclasses of TRL, TSA, TRV and TRT representing 0.5–1.0 mm, 1.0–1.5 mm and 1.5–2.0 mm and 2.0–2.5 mm diameter, higher root percentage in NLP than LP.

### Diversity pattern with Shannon-Weaver diversity index (H)

The comparative analysis of root morphology of the various groups showed clear differences for RSAs (Table 4). The genotypes were grouped into low, medium and high-performance categories based on the mean and standard deviation of each trait under both NLP and LP and then the percentile of genotypes in each category is derived over total number of genotypes (182). The majority of genotypes were grouped in to the medium performance category for all RSAs irrespective of P regimes. In the high-performance category, a more significant number of genotypes for all RSAs except PRL fell in the NLP regime, though the difference was very minimal. However, in the low-performance category, more number of genotypes fell in the LP regime for all the root traits except for PRL and RAD. In this category, the lowest percentage

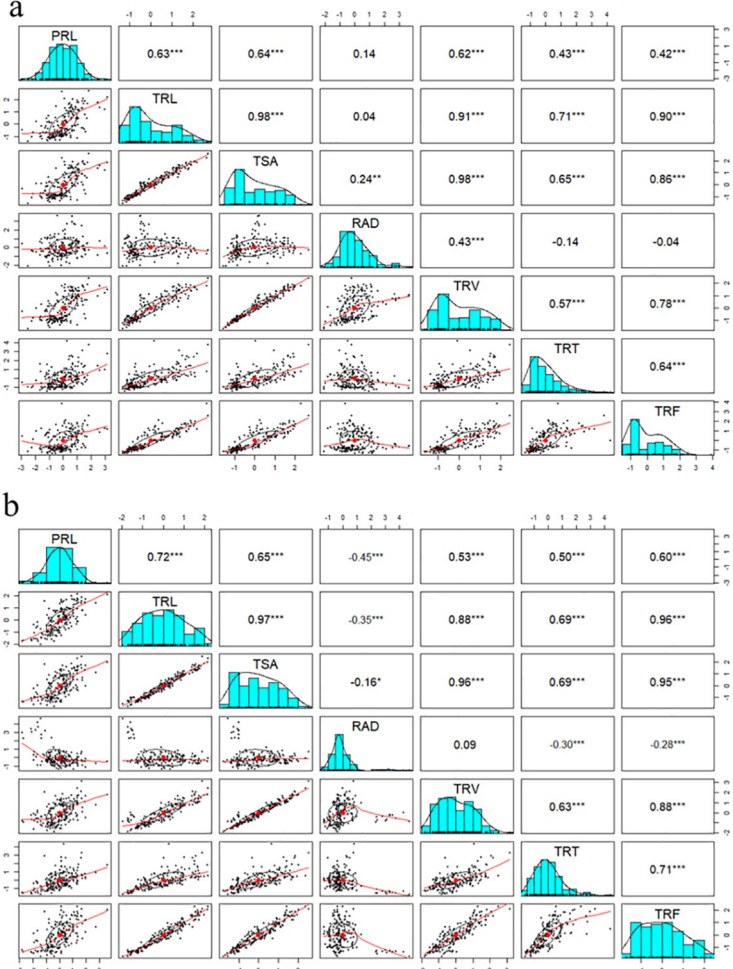

**Fig 2.** Relationship between measured root traits, their significance levels and distribution among wheat genotypes under non-limiting (NLP) (a) and limiting (LP) phosphorus (b) conditions.

of genotypes fell in LP were for PRL (12.64%) and RAD (4.40%). More interestingly, under LP conditions, more number of highest performing genotypes found for root traits, TRL (23.08%) and TSA (21.43%). The rate of genotypes falling in the low and high performing category have been presented in (Table 4).

The Shannon-Weaver diversity index (H') was calculated to investigate the diversity between the RSA traits in diverse wheat genotypes (Table 4). The H values for the PRL, TRL,

**Table 3. Root diameter intervals and respective percentage distribution of root traits of wheat genotypes under both non-limiting and limiting phosphorus conditions.**

| Traits | | TRL (%) | | | TSA (%) | | | TRV (%) | | | TRT (%) | | |
|---|---|---|---|---|---|---|---|---|---|---|---|---|---|
| Treatments | | Subclass | NLP | LP | Subclass | NLP | LP | Subclass | NLP | LP | Subclass | NLP | LP |
| Root diameter class (mm) | 0.0–0.5 | RL[1] | 82.36 | 87.66 | SA[1] | 55.27 | 64.68 | RV[1] | 28.22 | 35.04 | RT[1] | 98.35 | 99.00 |
| | 0.5–1.0 | RL[2] | 16.10 | 11.10 | SA[2] | 37.21 | 28.57 | RV[2] | 49.88 | 42.95 | RT[2] | 1.39 | 0.35 |
| | 1.0–1.5 | RL[3] | 1.12 | 0.89 | SA[3] | 4.78 | 4.24 | RV[3] | 11.64 | 11.48 | RT[3] | 0.06 | 0.04 |
| | 1.5–2.0 | RL[4] | 0.32 | 0.27 | SA[4] | 1.96 | 1.84 | RV[4] | 6.73 | 7.11 | RT[4] | 0.02 | 0.02 |
| | 2.0–2.5 | RL[5] | 0.10 | 0.08 | SA[5] | 0.78 | 0.67 | RV[5] | 3.51 | 3.3 | RT[5] | 0.01 | 0.01 |

**Table 4. Wheat genotypes performance classes (low, medium and high) and the diversity index (H') for the root system architectural traits under both non-limiting and limiting phosphorus conditions.**

| Traits | Treatments | Genotypes | | | Diversity index (H') |
|---|---|---|---|---|---|
| | | Low | Medium | High | |
| PRL | NLP | 28 (15.38) | 128 (70.33) | 26 (14.29) | **0.81** |
| | LP | 23 (12.64) | 128 (70.33) | 31 (17.03) | **0.81** |
| TRL | NLP | 25 (13.74) | 115 (63.19) | 42 (23.08) | **0.90** |
| | LP | 34 (18.68) | 119 (65.38) | 29 (15.93) | **0.88** |
| TSA | NLP | 27 (14.84) | 116 (63.74) | 39 (21.43) | **0.90** |
| | LP | 42 (23.08) | 105 (57.69) | 35 (19.23) | **0.97** |
| RAD | NLP | 13 (7.14) | 146 (80.22) | 23 (12.64) | **0.63** |
| | LP | 8 (4.40) | 161 (88.46) | 13 (7.14) | **0.43** |
| TRV | NLP | 29 (15.93) | 117 (64.29) | 36 (19.78) | **0.90** |
| | LP | 34 (18.68) | 116 (63.74) | 32 (17.58) | **0.91** |
| TRT | NLP | 13 (7.14) | 143 (78.57) | 26 (14.29) | **0.66** |
| | LP | 26 (14.29) | 134 (73.63) | 22 (12.09) | **0.76** |
| TRF | NLP | 21 (11.54) | 127 (69.78) | 34 (18.68) | **0.81** |
| | LP | 36 (19.78) | 114 (62.64) | 32 (17.58) | **0.92** |

Parenthesis: percent of individuals in each category.

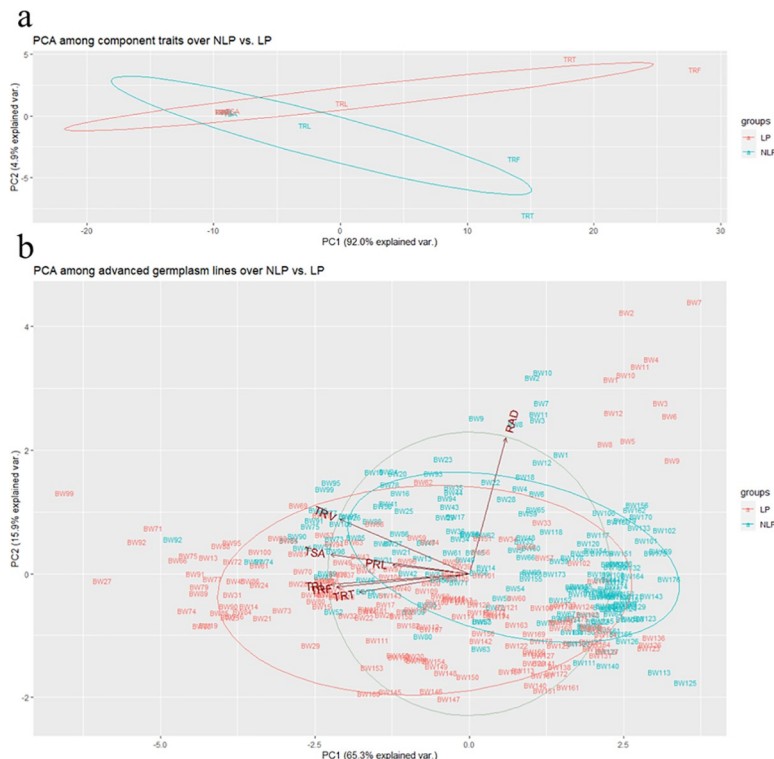

**Fig 3. Principal component analysis.** PCA for seven root architectural traits (a), PCA for genotypes under two P regimes. Green circle represents NLP and orange represents LP conditions (b). Proportion of variances for PC1 and PC2 are shown in parentheses.

TSA, TRV, RAD, TRT and TRF varied largely and overall it ranged from 0.43–0.97 across the roots traits. RAD, TRT and TRF showed relatively higher variation (0.63 vs 0.43; 0.66 vs 0.76 and 0.81 vs 0.92 for NLP vs LP conditions respectively). While, PRL did not vary at all under both the NLP and LP conditions. In comparison to NLP, a wide range of H' value i.e. 0.43 to 0.97 in LP, indicates that the expression pattern of these traits is highly influenced by availability of P. These traits were having the differential pattern of H' values over NLP and LP. In LP, higher H' values for all the traits except RAD (0.43) and TRT (0.79) were observed. In LP, the reduction in the H' value of RAD (0.43) indicates that this trait's full expression is hampered under limited P. In contrast, LP had given the ambience for higher expression of TRV, TRF and TSA with higher H' values i.e. 0.91, 0.92 and 0.97 respectively.

## Large amount of non-crossover interaction predicted by the principal component analysis

PCA based grouping of traits over LP and NLP conditions (Fig 3), indicated that the first principle component (PC1) had explained 92% variation defining the presence of a large amount of non-crossover interaction. The traits RAD, TSA, PRL and TRV were grouped together and having a low PC2 score. TRT, TRF and TRL were separated with different P regimes, in-between TRF and TRT had the high PC2 score followed by TRL in both P regimes. The distribution of genotypes with PCA over LP and NLP indicates that TRV, TRL, PRL, TRT, TRF and TSA are positively correlated, while RAD is not associated with these root traits. Among the tested wheat genotypes BW99, BW71, BW92, BW95, BW88, BW75, BW66, BW13, BW79, and BW89, were positively correlated with TRV, TRL, PRL, TRT, TRF and TSA with high trait expression value under LP. Meanwhile, genotypes BW92 and BW75 were having a high trait expression of these traits irrespective of P availability. The genotypes, BW2, BW10, BW7, BW8, BW11, BW3 in NLP; and BW2, BW7, BW4, BW11 in LP were having a positive association with RAD. While genotypes BW2, BW7, BW11 were common in both P regimes. 'Which Won Where' had given precise insights to select the best genotypes for respective P regimes. In LP, BW7 &, BW2 for RAD; BW99 for TRV, TSA, TRF, TRL; and BW27 for PRL, TRT were the best genotypes. While, BW10; BW92 for TRT, TRF, TRL; and BW52 for TSA, PRL, TRV were the best genotypes in NLP (Fig 4).

## Grouping wheat genotypes in to different classes based on comprehensive P response index (0–1)

The comprehensive response of all wheat genotypes for P use was estimated through a comprehensive synthetic index, using subordination function value analysis to enquire the response of root system architecture under limiting and non-limiting phosphorus conditions (S2 Table). Based on the comprehensive synthetic index, 182 wheat genotypes were grouped into five different classes which varied from 0.03 in 0.90 across the genotypes. First class comprised high P stress responsive genotypes with comprehensive P response index higher than 0.70. The variation in the RSAs of contrasting genotypes based on CPRI values is presented in Table 5 and Fig 5. Genotype BW181 with the highest CPRI value observed higher values for PRL, TRL, TSA, TRV, TRT, TRF and except for RAD under LP than NLP.

The second class comprised of 17 genotypes with CPRI values ranges between 0.50 and 0.70. A total of 44 genotypes fell in to class three which exhibited moderate responsive with CPRI values between 0.30 and 0.50 consists. Low P responsive genotypes (100) with CPRI values between 0.10 and 0.30 were grouped in to fourth class, while fifth class comprising 16 genotypes, including BW 9 and BW 139, poor performing genotypes for all LP traits, showed non-responsiveness with CPRI values below 0.10. Mean P response coefficient values for all

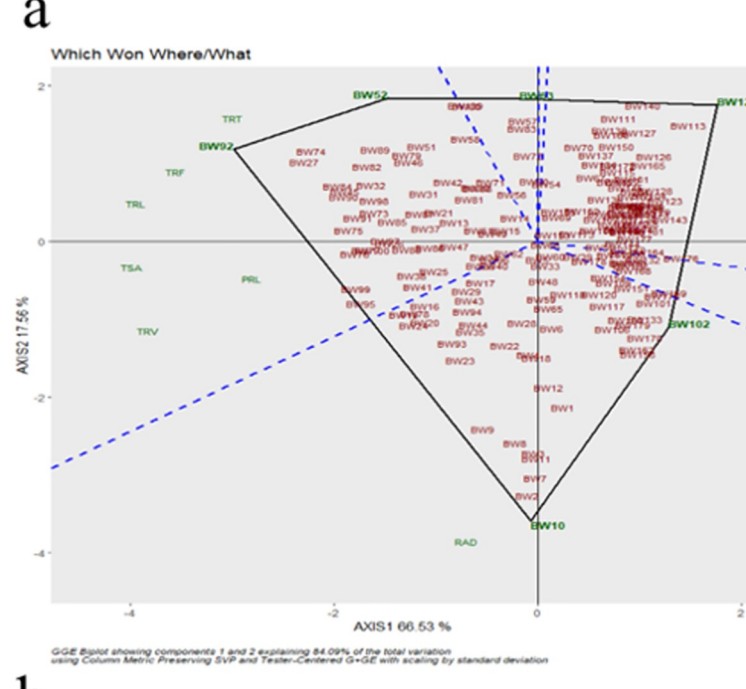

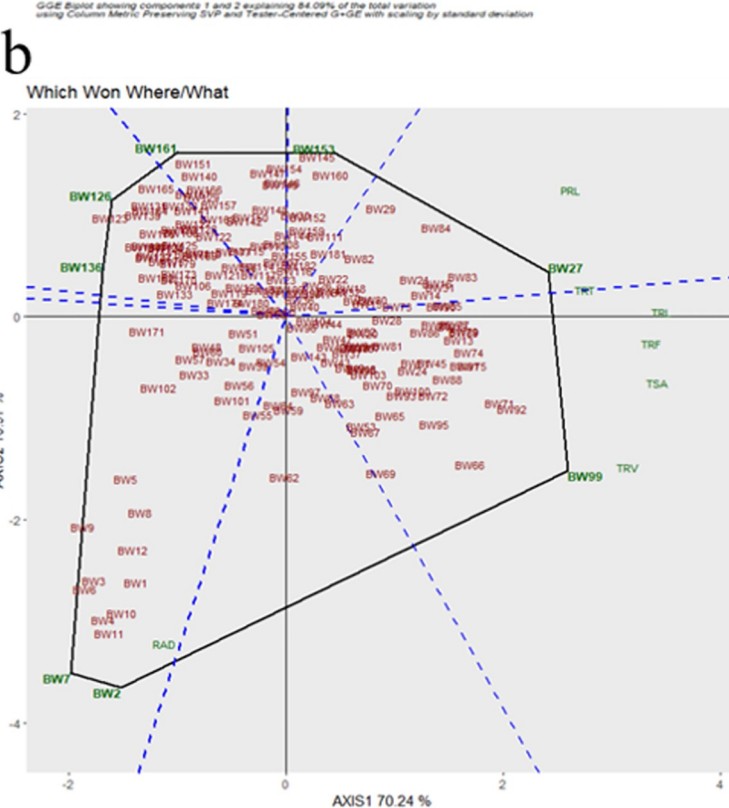

**Fig 4.** Which-won-where view of GGE biplot to show which genotype performed better for which trait under in (a) non-limiting and (b) limiting phosphorus conditions.

**Table 5. Comprehensive phosphorus response index values and top ten contrasting wheat genotypes for seven root system architectural traits under non-limiting and limiting phosphorus conditions.**

| GENOTYPES | CPRI | ROOT TRAITS | | | | | | | | | | | | | |
|---|---|---|---|---|---|---|---|---|---|---|---|---|---|---|---|
| | | PRL | | TRL | | TSA | | RAD | | TRV | | TRT | | TRF | |
| | | NLP | LP | NLP | LP | NLP | LP | NLP | LP | NLP | LP | NLP | LP | NLP | LP |
| Top 5 genotypes | | | | | | | | | | | | | | | |
| BW181 | 0.904 | 45.25 | 46.88 | 421.95 | 1095.46 | 38.89 | 98.91 | 0.31 | 0.29 | 0.30 | 0.71 | 1314.97 | 4399.64 | 1192.72 | 4537.69 |
| BW103 | 0.806 | 36.48 | 33.72 | 529.37 | 1416.75 | 49.83 | 128.96 | 0.30 | 0.29 | 0.38 | 0.93 | 1721.18 | 3996.07 | 1707.32 | 6391.79 |
| BW104 | 0.751 | 41.85 | 42.06 | 437.84 | 1141.38 | 40.38 | 107.31 | 0.29 | 0.30 | 0.30 | 0.80 | 1556.65 | 3888.83 | 1380.11 | 3807.77 |
| BW143 | 0.733 | 29.94 | 33.70 | 440.98 | 1140.43 | 43.02 | 105.90 | 0.31 | 0.30 | 0.34 | 0.78 | 1389.65 | 3260.42 | 1757.44 | 5883.29 |
| BW66 | 0.708 | 41.09 | 31.50 | 647.68 | 1654.60 | 66.41 | 159.78 | 0.33 | 0.31 | 0.54 | 1.23 | 2245.13 | 7375.77 | 3647.26 | 7438.71 |
| **% Change** | | **-0.34** | | **19.12** | | **18.25** | | **-0.40** | | **16.98** | | **20.97** | | **25.49** | |
| Bottom 5 genotypes | | | | | | | | | | | | | | | |
| BW9 | 0.033 | 47.19 | 21.33 | 699.40 | 434.54 | 87.32 | 51.20 | 0.39 | 0.39 | 0.87 | 0.49 | 3137.18 | 862.21 | 2267.88 | 1935.96 |
| BW139 | 0.039 | 43.81 | 34.31 | 767.76 | 559.12 | 73.82 | 48.90 | 0.31 | 0.28 | 0.57 | 0.34 | 8557.33 | 1895.12 | 2657.02 | 1977.49 |
| BW136 | 0.057 | 33.55 | 26.18 | 526.26 | 487.52 | 52.89 | 43.77 | 0.32 | 0.29 | 0.43 | 0.32 | 3857.57 | 1388.07 | 1920.56 | 1732.26 |
| BW3 | 0.060 | 46.75 | 25.22 | 585.75 | 430.61 | 73.40 | 54.34 | 0.39 | 0.42 | 0.73 | 0.56 | 1628.54 | 847.16 | 2170.52 | 1872.13 |
| BW8 | 0.074 | 49.68 | 29.86 | 635.72 | 533.48 | 78.77 | 63.73 | 0.39 | 0.39 | 0.78 | 0.62 | 2115.31 | 1617.29 | 1774.04 | 2141.80 |
| **% Change** | | **-19.81** | | **-12.35** | | **-14.76** | | **-1.31** | | **-16.43** | | **-30.71** | | **-4.62** | |
| HD 3226 | 0.15 | 54.53 | 48.17 | 1228.60 | 1499.68 | 139.09 | 159.65 | 0.36 | 0.34 | 1.25 | 1.35 | 2540.00 | 3429.33 | 4420.67 | 5730.00 |
| HDCSW18 | 0.16 | 56.50 | 50.60 | 1181.50 | 1608.41 | 121.04 | 146.99 | 0.33 | 0.29 | 0.99 | 1.07 | 3062.67 | 4406.00 | 4142.00 | 5002.00 |
| HD 2967 | 0.08 | 36.00 | 27.83 | 496.67 | 254.22 | 59.77 | 38.01 | 0.41 | 0.48 | 0.56 | 0.46 | 1519.33 | 451.33 | 1841.33 | 1012.67 |

root characteristics were highest for class 1, moderate for class 2, 3 and 4, and lowest for class 5, except for RAD. This result indicates that P efficient wheat genotypes with higher CPRI values also had higher P response coefficients.

The per cent change in the RSA traits from NLP to LP of the top five genotypes was positive for TRL (19.12), TSA (18.25), TRV (16.98), TRT (20.97), TRF (25.49) but PRL (-0.34) and RAD (-0.40) showed a negative trend. The highly responsive genotype class has shown that the reduction in PRL and RAD traits with increasing in RSA remaining traits under limiting phosphorous conditions. In case of bottom five genotypes with low P response, the per cent changes are PRL (-19.81), TRL (-12.35), TSA (-14.76), RAD (-1.31), TRV (-16.43), TRT (-30.71) and TRF (-4.62) which clearly evident that a decrease in the RSA traits in the LP conditions in comparison to the NLP condition for non-responsive genotypes. In order to have a comparative understanding, three high yielding commercial varieties viz., HD2967, HD3226 and HDCSW18 which were bred under optimum P environment were tested for the P response and they exhibited CPRI index value of 0.16, 0.15 and 0.08 respectively. These genotypes showed low P response, even though having a better root trait compared to most of the advanced breeding lines.

## Discussion

Phosphorus is a main component of nucleic acid, plant hormones, and largely defines crop growth, quality and consolidated yield [2, 3]. Insufficiency of P affects the photosynthetic ability and electron transport by repressing orthophosphate concentration in stroma, thus inhibiting ATP synthase activity [43]. Generally, greater root:shoot ratio was observed under P stress due to the tendency of plants to increase the root biomass to uptake more nutrients [44, 45] from the soil. The root system is a vital organ for the absorption of nutrients and water in plants and their rate of uptake is largely determined by physiological characteristics of plant

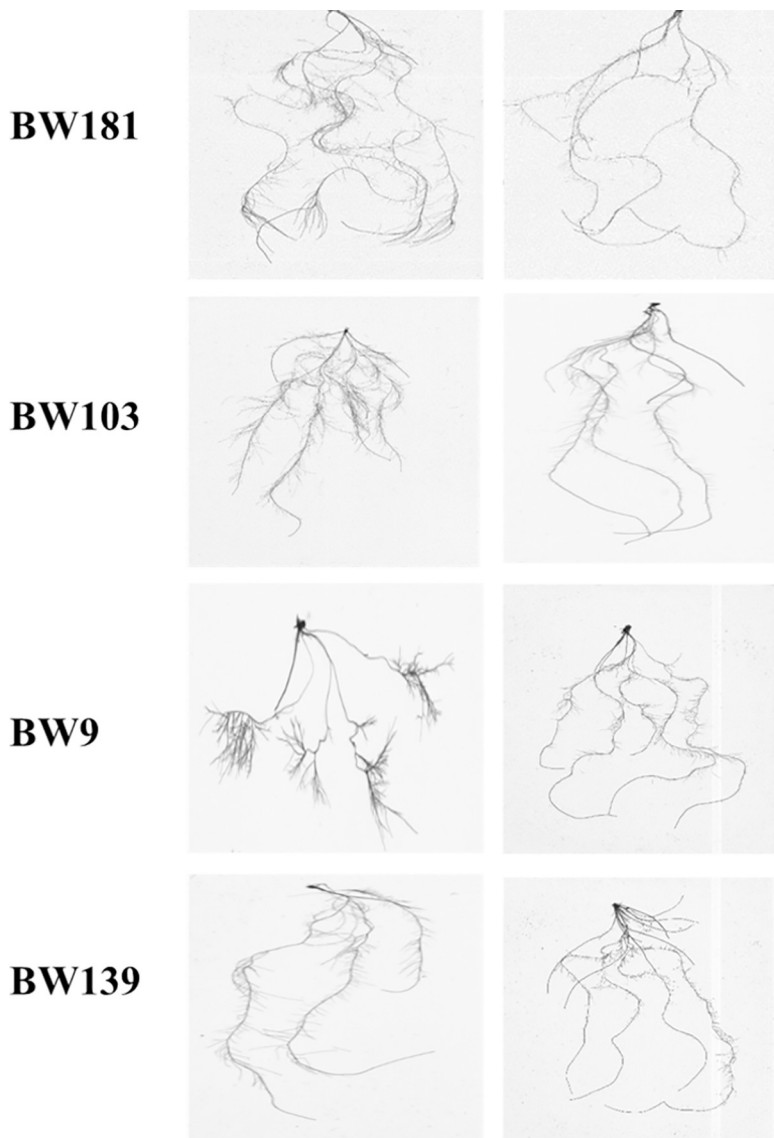

**Fig 5. Root system architecture of better performing (BW 181 and BW 103) and poor performing (BW 9 and BE 139) genotypes grown under non-limiting (NLP) and limiting (LP) phosphorus conditions.**

roots [17]. Therefore, the study of variation in root architectural traits and their associations with each other would help us to maximize the efficiency of nutrient and water use and identify key traits showing maximum positive response would help to improve wheat genotypes suitable for LP conditions. These traits would also be readily, transferred to high yielding, but P non responsive genotypes to develop higher yielding P responsive cultivars for the future world.

Though understanding the plants root system is very important to estimate nutrient absorption efficiency and to characterise, and select better genotypes responsive to P uptake, there are no systematic studies undertaken so far in wheat crop on Indian breeding material. Hence, the current investigation was carried out to systematically characterize Indian wheat germplasm for root architectural traits and to understand their differential ability to phosphorus use efficiency and response.

Analysis of variance revealed the presence of significant variations for genotypes, P regimes, and G×P interactions for all the root traits under present study. The significance of G×P indicates that wheat genotypes root traits have been substantially modified by conditions LP and NLP under the hydroponics. This indicates that the tested genotypes possess differential responsive ability and they adopt different root traits for available P conditions. The maximum mean sum of squares among the RSA traits was observed for TRT, not only across genotypes and the P regimes, but also for their interaction (G×P). This suggests that there is an ample scope for exploring this trait to improve wheat genotypes for better P use efficiency. The low CV, for RAD established the fact that, the tested genotypes performed stably across the replications, leading error free estimations. But, TRT has showed little high variation, as these root traits may vary considerably within the common treatment. This makes interpretation through root studies, more complicated and unrealistic in developing a standard scale value. The, varied response indicated for these root traits creates a definitive platform for the adoption of unique selection and breeding strategies for developing the P responsive genotypes in wheat. High broad-sense heritability of the RSAs gives indications of the presence of underlying genomic regions with significant genotypic effect on these root traits. So, there is always a scope for improving wheat breeding lines, through selection and introgression of underlying gene(s) responsive to P use without much influence of environment. The genetic stability of a particular genotype for any trait under consideration is determined by low coefficient of variation and high heritability of respective traits [42].

The, reduction of mean PRL and enhancement in TRL of genotypes under LP our findings is also supported by the change in primary root elongation reported as a typical phenomenon in response to P deprivation in other major cereal rice (*O. Sativa*) [46, 47]. The enhanced primary root length growth was reported in rice [48] and green gram [42], but was found to be inhibited in Arabidopsis in response to P deficiency [49]. The reduction of cell differentiation within the primary root meristem and the inhibition of cell proliferation in the root elongation zone reduce the PRL in P deficient [50, 51]. The inhibition of primary root growth and proliferation of lateral root formation is enhanced by relocating mitotic activity at the sites of lateral root formation by P deficiency, resulting in increased total root length [52, 53]. Likewise, the reduction of RAD in LP in our finding is in line with previous findings, as a 30% reduction in root diameter from high P to low P [54]. A reduction in root diameter, root mass density, and an increase in specific root length was observed in temperate pasture species [55]. In comparison to adequate P condition, root diameter reduction was recorded under low P in Zea mays [56] and *Aegilops tauschii* [57] under P limitation. Earlier researchers have observed an higher per cent increment in root length by 113% and 80%, respectively, in two wheat cultivars, Crac and Tukan with a reduction in root diameter [58]. Also P stress, showed greater root surface area and root volume in the P efficient genotypes of mungbean [11].

Association studies helps to identify highly interactive traits, in response to a given treatment and hence, helps in picking positive traits for selection and improvement. In both P regimes, all RSA traits viz., TRV, TRL, PRL, TRT, TRF and TSA were having the moderate to high associations with each other except for RAD. However, the correlations of RAD with other traits were changed with P regimes, indicating influence of availability of P has substantially affected the way these traits interact. Under non-limiting conditions also TRL was positively correlated with TSA and TRV in wheat [59] showing their noninfluential interactive modifications under different P regimes. In LP, RAD had significant negative correlations with PRL, TRL, TSA, TRT and TRF and no correlation with TRV. In contrary, under NLP the significant positive correlation of RAD with TRV and TSA and a non-significant association with other traits were observed. This proves the fact that, under stressed environment of P unavailability RAD reduces drastically independent of other root traits where they show increased mean values. Therefore,

RAD is a key characteristic feature for differentiating tested root traits and to study of the impact of available P on plant growth. The vigorous root system of the plant thus not only encourages the establishment of good crops but also ensures the survival of the plants under these stressful conditions. Moreover, at the plant's early growth stage, the absorption of nutrients is facilitated by vigorous root growth along with higher root length and surface area [60].

Even though the genetic variation of the root system differs from plant to plant, the existence of very fine to fine roots determines the most percentage of root traits, which is essential for water and nutrient uptake [61, 62]. In this study, we observed the high percentage of fine roots in diameter class from 0.0 to 0.5 mm in LP compared to NLP condition, while the percentage of roots with >0.5 mm diameter was more in NLP condition. It was very interesting to note that, two root traits viz., TRL and TRT had a maximum percent of very fine roots (0.0–0.50mm) under both the conditions, with a higher mean trending towards LP. This showed the deciding factor of P availability on the percent of fine root distribution pattern at different diameter classes in the wheat genotypes. Due to fine roots tend to turn over more rapidly than coarse roots, the carbon cost of producing finer roots may be higher as these will have to be replaced more frequently [63]. Under P deficient condition, plant species produce more fine roots that increase total root area in contact with larger soil volume per unit of root surface area, thereby increasing P uptake rates [22, 63]. Enhanced fine root production results in improving the total adsorption area, as an adaptive strategy of crop plants stress-tolerance mechanism [64]. In addition, the presence of fine root hairs [65] and roots with less diameter allows better absorption of nutrients due to increased available absorptive surface area. This must prove an obviousness that, the genotypes with the better root systems outperform in terms of yield as compared to genotypes with poor root traits.

Based on mean performance and standard deviation of root characteristics as a criterion of selection [41] wheat genotypes were classified into high, medium and low- efficiency classes. In the high-performance group, a high percentage of genotypes for RSAs, especially TRL and RAD in the NLP regime than LP, indicate that only a few selective genotypes remain responsive in limited P and the genotypes itself vary in their response to P. As expected, the high percentage of genotypes for RSAs except PRL and RAD fell in LP under the low-performance group. The Shannon-Weaver diversity index (H') revealed high diversity among the genotypes for RSAs over NLP and LP. The presence of a wide range of H' value i.e. 0.43 to 0.97 in LP indicates that the expression pattern of RSAs is highly influenced by the level of P. The higher H' value indicates balanced frequency distribution and greater phenotypic diversity, while lower H' value indicates an extremely unstable frequency distribution with a lack of diversity for that trait [41]. Regardless of root length, larger diversity in root diameter is due to the changes in the fine root distribution pattern for each root diameter in response to the nutritional environment [61]. In contrast, LP had given the ambiance for higher expression of TRV, TRF and TSA with higher H' values. The Shannon-Weaver Diversity Index (H') were previously used to describe the diversity of root traits in rice [66], maize [67], and mungbean [42].

The PCA analysis to study the expression pattern of root traits over LP and NLP revealed that a high PC score (92%) defines the presence of a large amount of non-crossover interaction. The traits, namely RAD, TSA, PRL and TRV had a PC2 score close to zero, which shows the parallel response of these traits for a large number of genotypes in either direction over LP and NLP (Fig 2). The traits namely TRV, TRL, PRL, TRT, TRF and TSA were positively correlated with each other, and RAD was independent of other traits. Genotypes pertaining to LP, with high trait expression value were positively correlated with TRV, TRL, PRL, TRT, TRF and TSA. The same correlation pattern among the traits was observed in mungbean [42]. The genotypes BW92 and BW74 had a high trait expression, and BW123 and BW126 were having a low trait expression for these traits irrespective of P regimes. Therefore, genotypes having

high and low performance can be utilized for making bi-parental populations that would have a very low influence on P regimes. With the help of the 'Which Won Where' biplot best genotypes for individual traits in both P regimes were selected (Fig 3). Our results suggested that TRL, TSA, TRV and TRF were sufficient to explain the most of variation and these were proved to be ideal traits for phosphorus uptake efficiency screening at seedling stage. As most of these traits showed very close relatedness in terms of explaining the variation we, can choose TRL as the main trait as it is a combination of many component RSA traits determining the efficiency of genotypes.

CPRI is based on relative trait values i.e. PRC of traits and degree of membership between trait value and P response i.e. MFVP [68], considering all traits together regardless of the nature of the trait. It represents the response of the genotypes based on the magnitude of change of all traits from NLP to LP. The genotypes BW181, BW103, BW104, BW143, and BW66 were found to be most P responsive based on CPRI value. The most contributing in top five P stress-responsive genotypes was TRF followed by TRT, TRL, TSA and TRV. Total root length and root dry weight were able to provide the most contribution to total variation and sufficient to improve other root traits in maize [25, 49]. The PRL and RAD have classified the least contributing traits in gaining the P responsive due to their average negligible share in CPRI index of the top five P responsive genotypes. In contrast, BW9, BW139, BW136, BW3 and BW8 were classified as non-responsive genotypes. The drastic reduction was seen in TRT followed by PRL, TRV, TSA and TRL. The least affected trait in the bottom five genotypes was RAD followed by TRF. This result indicates that RAD is the least important trait in developing the P efficiency of genotypes, while TRF is of very high importance in increasing P limitation efficiency. However, TRF and RAD were not reduced drastically in limited P, which means both the traits are essential in the survival of the plants in limited P. This classification is important for selection of genotypes for desirable RSA traits under limiting p conditions. Furthermore, these genotypes with contrasting root traits can be used in recombination breeding programme to develop P efficient cultivars [69, 70]. The three commercial varieties, used for a comparative understanding of P response, showed very low comprehensive phosphorus response index values, as they were initially bred under non-limiting P conditions. In our findings, HDCSW18 was identified as low P responsive because it is having high TRL, PRL, TSA, TRF and TRT values in both P regimes. That means this variety might be having an inherent capacity to show constant response to changing P regimes for RSA traits. Under CA environments the nutrient use efficiency is low as compared to normal tillage conditions. Because the applied fertiliser is retained for longer time on the soil surface and the plants should have the inherent capacity to absorb the nutrients from the soil rather than totally dependent on the external supply of fertiliser. HDCSW18 has been bred under Conservation Agriculture (CA) environments and therefore believed to be having better absorption capacity for the nutrients due to its strong RSA traits. Based on the pedigree information, few genotypes which shared common parentage varied for their P response index. For example, genotypes BW143, BW69, BW1, BW162, BW 6 and BW 171 developed through a cross CSW2/ HD2932+Yr15, showed high to very low/non-responsiveness. This may be due to the reason that, both the parents are contrasting for P use efficiency. It also shows that the P use efficiency is polygenic in nature and these lines may be suggestive to be used as parents in developing mapping populations for dissecting P use efficiency.

## Conclusion

To reduce the P footprint on the environment and to lessen the economic burden due to over P fertilisation, increasing the PUE of the important cereal crop like wheat is the present-day

need. Knowing the complexity of root traits exploration under field conditions, its labour intensiveness and the large influence by the environment these traits, these traits are least explored by the breeders in improving crop plants. Our earlier findings on nitrogen use efficiency, another important mineral, revealed a significant association between hydroponic and soil-filled pot conditions for nitrogen use efficiency. Therefore, screening of RSAs under hydroponics was used to measure the P use response of wheat genotypes. Hydroponics helps in the preliminary screening of a large number of genotypes with minimum efforts under controlled conditions. Our results indicated that the P-efficient genotypes showed increased fine root anatomy and their adaptation strategy to P limitation conditions. Under LP-condition, important RSA traits like TRL, TSA, TRV, TRT and TRF of P-efficient genotypes were significantly higher than P-inefficient and these traits can be used as direct measures of P use response and can be used readily in selecting genotypes with higher P efficiency. We could identify five genotypes BW 181, BW 103, BW 104, BW 143 and BW 66, with higher P response and further these genotypes can be used for developing P efficient wheat cultivars. The identified highly efficient genotypes can also be used in developing mapping populations with ineffective genotypes for dissecting the genomic regions responsible for PUE. It is further needed to investigate the relationship of RSA in the adult plant stage and their correlation with the seedling root traits for a better understanding of the differential expression of these traits in response to natural P availability conditions.

## Supporting information

**S1 Table. List of 182 wheat genotypes used in the study.**
(PDF)

**S2 Table. Comprehensive phosphorus response index value of 182 wheat advanced breeding lines used in the study.**
(PDF)

## Acknowledgments

Authors are thankful to Dr. A. K. Singh, Director, IARI, New Delhi, Dr. Rajbir Yadav, Head, Division of Genetics, Dr. Vinod, Professor Division of Genetics, IARI, New Delhi and Dr. Akshay Talukdar, Principal Scientist, Division of Genetics, IARI, New Delhi (Incharge, National Phytotron Facility) for providing the necessary facilities for smooth conductance of research.

## Author Contributions

**Conceptualization:** Rajbir Yadav.

**Formal analysis:** Palaparthi Dharmateja, Manjeet Kumar, Pranab Kumar Mandal.

**Funding acquisition:** Rajbir Yadav.

**Investigation:** Palaparthi Dharmateja, Kamre Kranthi kumar, Narain Dhar, Rihan Ansari, Nasreen Saifi.

**Methodology:** Palaparthi Dharmateja, Rakesh Pandey, Rajbir Yadav.

**Project administration:** Rajbir Yadav.

**Resources:** Rakesh Pandey, Rajbir Yadav.

**Software:** Palaparthi Dharmateja, Manjeet Kumar, Prashanth Babu, Vipin Tomar.

**Supervision:** Palaparthi Dharmateja, Manjeet Kumar, Pranab Kumar Mandal.

**Validation:** Palaparthi Dharmateja.

**Visualization:** Rakesh Pandey, Naresh Kumar Bainsla, Kiran B. Gaikwad, Rajbir Yadav.

**Writing – original draft:** Palaparthi Dharmateja, Manjeet Kumar.

**Writing – review & editing:** Manjeet Kumar, Prashanth Babu, Rajbir Yadav.

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
