## [Decision Letter · Decision Letter 0]

18 May 2021

PONE-D-21-09696

Deciphering the change in root system architectural traits under limiting and non-limiting phosphorus in Indian bread wheat germplasm

PLOS ONE

Dear Dr. Yadav,

Thank you for submitting your manuscript to PLOS ONE. After careful consideration, we feel that it has merit but does not fully meet PLOS ONE’s publication criteria as it currently stands. Therefore, we invite you to submit a revised version of the manuscript that addresses the points raised during the review process.

Please revise the manuscript  and resubmit at the earliest. 

We look forward to receiving your revised manuscript.

Kind regards,

Reyazul Rouf Mir, PhD

SKUAST-Kashmir

Academic Editor

PLOS ONE

Journal Requirements:

3.Thank you for stating the following financial disclosure:

 "NO"

Additional Editor Comments:

The manuscript was reviewed by two reviewers and both have recommended revision of the manuscript before its acceptance. Therefore, you are requested to address all the comments of the reviewers and resubmit manuscript at the earliest for further processing at our end.

Reviewers' comments:

Reviewer's Responses to Questions

**Comments to the Author**

1. Is the manuscript technically sound, and do the data support the conclusions?

Reviewer #1: Yes

Reviewer #2: Yes

2. Has the statistical analysis been performed appropriately and rigorously? 

Reviewer #1: Yes

Reviewer #2: Yes

3. Have the authors made all data underlying the findings in their manuscript fully available?

Reviewer #1: Yes

Reviewer #2: Yes

4. Is the manuscript presented in an intelligible fashion and written in standard English?

Reviewer #1: Yes

Reviewer #2: Yes

5. Review Comments to the Author

Reviewer #1: Introduction:

The authors have presented the information for wheat crop with respect to the nutrient use. However, very little information on root traits is presented which needs improvement. Authors should include relevant information which root traits have been identified in wheat or other cereal crops particularly rice for efficient uptake and utilization of P. Few statements are written without appropriate citation.

Material and Methods:

A total of 182 lines were used in this study which is very big set for this study. I wish to now if any commercial variety have been included in this study. If yes, then comparisons should be have been made against that as this will give information how the other genotypes are behaving in comparison to that commercial one. The genotypes having better root traits than that of commercial should have been selected (less affected traits under limited P).

The methodology and statistical analysis is appropriate for the study.

Results:

This is very basic study conducted to evaluate the genotypes for response to P.

The results are appropriately described as per the study.

However, no shoot biomass information is included. I will be happy if the authors should include the shoot biomass in this study also to get information about the root/shoot biomass ratio among different genotypes.

Why the primary root length is affected under limited P environment needs justification. Like the genotypes are diverting energy towards only root traits or also to sustain the biomass. The biomass will be converted into final yield.

The results needs to be presented in comparison to best commercial genotype used in the study.

Discussion:

The discussion part is mostly the repetition of results. The authors need to defend their results in relation to the studies conducted on root traits in wheat or related cereals. The discussion part needs to be thoroughly revised for better understanding and defending the results related to the identification of P efficient genotypes.

How the authors will justify that the genotypes which are having good root traits will suffer less in terms of yield as compared to other having poor under limited P. Only those genotypes will be beneficial to be utilized which perform better/ comparable under both limited and non-limited P conditions with respect to yield.

Reviewer #2: General comment:

The work related to Root System Architecture (RSA) under Phosphorous limited conditions in wheat has not been studied comprehensively due to its difficulty in phenotyping. The manuscript reports identification of RSAtraits and genotypesinfluencingPuse efficiency under hydroponics conditions, which I feel a reliable strategy and could be used effectively for screening.These P efficient genotypes, as they are advanced and fixed lines, could serve as excellent donors in developing P efficient wheat genotypes. However, the manuscript need improvement especially in discussion part. I would suggest moderate revision for this manuscript. Specific comments are as follows.

Introduction

Introduction is well written. The importance of P in improving the productivity is well explained. However, authors should explain more about in RSA traits under P efficient and P limited conditions. If studies are limited in Wheat,other cereal crops may be explored like maize, rice etc. In rice, good number of papers are available on RSA traits influenced by P use efficiency (e.g. The role of root size versus root efficiency in phosphorus acquisition in rice. Mori et al 2013 Journal of Experimental Botany,2016, 67:1179–1189).

Material and Methods

• 182 advanced breeding lines comprising of diverse parentages is a good experimental material for this study.

• Statistical analysis is appropriate.

• Comprehensive phosphorus response index value is well calculated and is important identifying P responsive and non-responsive genotypes.

• Authors have not included released varieties in the experimental material Why?

Result

• The results are well presented.

• Subtitles should not be lengthy. The authors have written important findings in the subtitles which I feel quite good however, it will be confusing for the readers because of their lengthiness. Try to make it concise where it is possible or give name of the analysis as subtitle.

• In PCA, Figure 3B. the angle of eigen vectors between TRV, TRL, PRL, TRT, TRF and TSA is less than 90o, it indicates they are quite related to each other and we can take most important trait for the study leaving other less important traits. Which trait do you think is more important under limited P conditions and you advocate in the abstract for the readers?

• In title of figure 5 for better clarity, rather than writing RSA of contrasting genotypes, write better performing (BW 181 and BW 103) and poor performing (BW 9 and BE 139) genotypes grown under non-limiting (NLP) and limiting (LP) phosphorus conditions

• In P efficient genotypes, viz., BW181, BW103, and BW66 a common parent noticed is HDCSW 18. This is a very high yielding wheat variety developed by IARI, New Delhi for conservation agriculture conditions. It would be great if authors could write more about this variety in relation to its P or N use efficiency in results as well as in discussion.

• Similarly, in the parentage of P non-efficient wheat genotypes like BW 9, BW 139, BW 3, two mega wheat varieties HD 2967 and HD 2733 are involved. If authors would come out with a reason why recombinants of these best genotypes adapted to farmers field showed poor P efficiency? If authors would have involved released varieties (popular varieties like HDCSW 18, HD 2967, HD 3086, HD 2733 etc.) in this study, it would have been good information.

Discussion

• Discussion is quite lengthy and need to be revised.

• Results are repeated in the discussion which is not required.

• Authors can cite references from other cereal crops pertaining to RSA under different nutrient stresses.

• In the discussion, author may give statements on genetics of P use efficiency in studied genotypes based on their parentages e.g.

Six genotypes shared common parentage (CSW2/ HD2932+Yr15); however, they differ for their P response index. BW143 is highly responsive, BW69 is moderately responsive, BW1, BW162, and BW 171are low responsive and BW 6 is non-responsive.

Nine genotypes having common parentage (HD2967/HD2887//HD2946/HD2733) showed varied P response index. BW70, BW158 are responsive; BW174, BW155 are moderately responsive; BW124, BW167, BW151 are low responsive and BW139, BW2 are non-responsive.

Five genotypes with parentage CSW3/HD2932+Yr10 showed varied P response index. BW 116 is responsive; BW 169 is moderately responsive; BW92, BW106 are low responsive; and BW136 is non-responsive.

6. PLOS authors have the option to publish the peer review history of their article (what does this mean?). If published, this will include your full peer review and any attached files.

Reviewer #1: **Yes: **Vikas Gupta

ICAR-Indian Institute of Wheat and Barley Research Karnal, Haryana (INDIA)

Reviewer #2: **Yes: **Sundeep Kumar

---

## [Author Response · Author response to Decision Letter 0]

28 Jun 2021

Dear Editor,

We have addressed all the comments of the reviewers and resubmitted the manuscript with appropriate changes and additional information for making the manuscript more plausible. Thank you.

With best regards,

Authors

---

## [Decision Letter · Decision Letter 1]

26 Jul 2021

Deciphering the change in root system architectural traits under limiting and non-limiting phosphorus in Indian bread wheat germplasm

PONE-D-21-09696R1

Dear Dr. Yadav,

We’re pleased to inform you that your manuscript has been judged scientifically suitable for publication and will be formally accepted for publication once it meets all outstanding technical requirements.

Kind regards,

Reyazul Rouf Mir, PhD

Academic Editor

PLOS ONE

Additional Editor Comments (optional):

The Manuscript has been revised in line with the comments made by the reviewers. Therefore may be accepted now for publication.

Reviewers' comments:

Reviewer's Responses to Questions

**Comments to the Author**

1. If the authors have adequately addressed your comments raised in a previous round of review and you feel that this manuscript is now acceptable for publication, you may indicate that here to bypass the “Comments to the Author” section, enter your conflict of interest statement in the “Confidential to Editor” section, and submit your "Accept" recommendation.

Reviewer #1: All comments have been addressed

2. Is the manuscript technically sound, and do the data support the conclusions?

Reviewer #1: Yes

3. Has the statistical analysis been performed appropriately and rigorously? 

Reviewer #1: Yes

4. Have the authors made all data underlying the findings in their manuscript fully available?

Reviewer #1: Yes

5. Is the manuscript presented in an intelligible fashion and written in standard English?

Reviewer #1: Yes

6. Review Comments to the Author

Reviewer #1: Authors have addressed all the queries raised and included suggestions in the revised manuscript. Good information have been generated in the present study related to the root system architecture under low P availability. The genotypes W 181, BW 103, BW 104, BW 143 and BW 66 identified in the present study will be useful for improving the P use efficiency.

7. PLOS authors have the option to publish the peer review history of their article (what does this mean?). If published, this will include your full peer review and any attached files.

Reviewer #1: **Yes: **Vikas Gupta, ICAR-Indian Institute of Wheat and Barley Research, Karnal (INDIA)

---

## [Editor Report · Acceptance letter]

27 Aug 2021

PONE-D-21-09696R1 

Deciphering the change in root system architectural traits under limiting and non-limiting phosphorus in Indian bread wheat germplasm 

Dear Dr. Yadav:

I'm pleased to inform you that your manuscript has been deemed suitable for publication in PLOS ONE. Congratulations! Your manuscript is now with our production department. 

Kind regards, 

on behalf of

Dr. Reyazul Rouf Mir 

Academic Editor

PLOS ONE